# Integrated Multi-Omics Analysis Uncovers Immune–Metabolic Interplay in Hepatocellular Carcinoma Tumor Microenvironment

**DOI:** 10.3390/cancers17213565

**Published:** 2025-11-03

**Authors:** Jong-Heon Park, Dae Won Sim, Sook-Young Kim, Joon Young Choi, Seung Hyup Hyun, Je-Gun Joung

**Affiliations:** 1Department of Life Sciences, Graduate School, CHA University, Seongnam 13488, Gyeonggi-do, Republic of Korea; pjongheon97@chauniv.ac.kr (J.-H.P.); tlaeodnjs34@chauniv.ac.kr (D.W.S.); 2CHA Research Institute (CHARI), CHA Bundang Medical Center, Seongnam 13488, Gyeonggi-do, Republic of Korea; skylove012@chauniv.ac.kr; 3Department of Nuclear Medicine, Samsung Medical Center, Sungkyunkwan University School of Medicine, Seoul 06351, Republic of Korea; jynm.choi@samsung.com; 4Department of Biomedical Sciences, CHA University, Seongnam 13488, Gyeonggi-do, Republic of Korea

**Keywords:** immune activity score, epigenetic regulation, transcriptome, prognosis markers

## Abstract

This study addresses the urgent need to better understand hepatocellular carcinoma (HCC), a highly lethal cancer, by analyzing the tumor microenvironment (TME). Using multiple omics profiling of HCC patients, we classified tumors by immune activity and discovered significant metabolic and immune-related genes (*AGXT2*, *DPYS*, and *TNFSF8*) whose expression is epigenetically regulated. Single-cell RNA sequencing revealed the cell type-specific roles of these genes, which link higher expression with improved prognosis. By elucidating the intersection of metabolic, immune, and epigenetic regulation, this study offers insights to propose new therapeutic targets and advance precise therapeutic strategies in HCC.

## 1. Introduction

Hepatocellular carcinoma (HCC) is the most common primary liver malignancy worldwide, accounting for 75–85% of all liver cancers and contributing substantially to cancer-related mortality [1]. The major etiologies of HCC include chronic infection with hepatitis B or C virus, alcoholic liver disease, and metabolic disorders such as NAFLD and NASH. Moreover, drug-induced liver injury (DILI) and chronic metabolic dysfunction-associated liver disease are recognized as growing contributors to HCC development, all of which induce persistent inflammation and hepatic injury that ultimately lead to genetic mutations and cirrhosis. Notably, cirrhosis is a key stage in HCC pathogenesis and is accompanied by hallmark features such as vascular invasion, increased cellular proliferation, and immune evasion during tumor development [2,3,4].

Given the complex pathogenesis of HCC, various diagnostic modalities are currently employed for HCC, including blood-based tumor markers such as alpha-fetoprotein (AFP) and circulating tumor DNA (ctDNA) analysis, as well as imaging techniques like ultrasound and PET/CT [5,6]. Previous transcriptomic studies have identified genes involved in key metabolic pathways associated with FDG uptake, demonstrating their prognostic significance and clinical applicability. However, single transcriptome-based analyses have limited capacity to comprehensively capture the molecular complexity of HCC, including gene mutations, protein expression changes, metabolic alterations, and DNA methylation.

To address these limitations, recent research has focused on multi-omics approaches that integrate diverse molecular data, including genomic, proteomic, metabolomic, and epigenomic profiles [7]. Multi-omics studies facilitate the simultaneous identification of a broad spectrum of molecular changes such as genetic mutations, protein expression dynamics, metabolic pathway alterations, and DNA methylation patterns [8], which are complexities that are often missed by transcriptome analysis alone. This integrated strategy expands opportunities for predicting clinical outcomes, identifying novel biomarkers, and developing tailored therapies. Notably, DNA methylation plays a pivotal role in shaping the complexity of the HCC tumor microenvironment (TME) by directly regulating the expression of cancer-related genes, including tumor suppressors and oncogenes [9,10,11].

Our study enhances prior HCC multi-omics research by integrating immune and metabolic features across genomic, transcriptomic, and epigenetic layers to delineate clinically relevant TME characteristics and identify potential therapeutic targets, whereas previous studies predominantly focused on molecular subtyping, mutation patterns, or immune profiling alone. In this study, we performed integrated analyses of transcriptomic, epigenomic, and single-cell transcriptomic data from HCC patient samples to identify genes influencing immune and metabolic functions in the TME. We further characterized genes regulated by epigenetic mechanisms in different microenvironments and assessed their association with clinical prognosis. Taken together, the identification of epigenetically regulated transcriptomic markers highlights the importance of multi-omics approaches for deciphering the complexity and heterogeneity of the HCC TME.

## 2. Materials and Methods

### 2.1. Patients and Clinical Information in the Study Cohort

Tissue samples were obtained from patients who underwent treatment and surgery for hepatocellular carcinoma between May 2009 and August 2015. Tumor samples (*n* = 60) and paired normal tissue samples (*n* = 60) were collected after surgical resection and prior to any radiation or chemotherapy, and only those classified as Edmondson-Steiner grade II by pathological diagnosis were included. The samples were provided by the Samsung Medical Center Biobank, collected in accordance with guidelines established by the Ethics Committee of Samsung Medical Center (IRB #2017-04-022). Since this study used data from a Biobank, the requirement for written informed consent was waived.

### 2.2. RNA Sequencing and Methylation Array

Whole-transcriptome sequencing was performed using libraries constructed from 500 ng of total RNA extracted from each sample using the Qiagen RNeasy Mini Kit (QIAGEN Inc., Valencia, CA, USA). Library construction was carried out using the Illumina TruSeq RNA Sample preparation v2 (Illumina, San Diego, CA, USA), following the manufacturer’s protocols. Sequencing was performed on a HiSeq 2500 with TruSeq Rapid PE Cluster Kit (Illumina, San Diego, CA, USA) and the TruSeq Rapid SBS Kit (Illumina, San Diego, CA, USA) generating 100 bp paired-end reads. The resulting FASTQ files were aligned to the hg19 human reference genome using TopHat version 2.0.6. Raw read counts were then obtained from the aligned BAM files using HTSeq version 0.6.1. Normalization of raw read counts was conducted using the functions implemented in the DESeq2 package. Gene expression fold changes were calculated by dividing the gene expression in tumor tissue by that in normal liver tissue, and values were log_2_-transformed for normalization.

Genomic DNA samples were subjected to bisulfite conversion using the Zymo EZ DNA Methylation Kit according to the manufacturer’s instructions. The converted DNA was then amplified before hybridization to the Illumina Infinium MethylationEPIC BeadChip. Following hybridization, chips were stained and scanned to generate fluorescent signals, which were subsequently processed to produce IDAT files for downstream methylation analysis. IDAT files were generated using the Illumina Infinium MethylationEPIC BeadChip Kits (Illumina Inc., San Diego, CA, USA) platform. Each methylation data point was represented as a fluorescence signal generated by methylated (*M*) and unmethylated (*U*) alleles. The ratio of the fluorescence signals was calculated as $\beta =\frac{max(M,0)}{U+M+100}$. The calculated *β*-value represents the methylation level of each CpG site, ranging from 0 to 1, corresponding to a methylation percentage from 0% to 100%.

### 2.3. Consensus Matrix

Immune scores obtained from all four algorithms were aggregated. Hierarchical clustering of the scaled immune scores was performed using the ConsensusClusterPlus R package (v1.66.0) (https://bioconductor.org/packages/release/bioc/html/ConsensusClusterPlus.html). Clustering parameters included 100% sample usage, 1000 replicates, and consensus classification into three clusters. The Ward. D linkage method was applied to minimize intra-cluster variance.

### 2.4. Whole-Exome Sequencing

The gDNA of the samples was sheared with an S220 ultra-sonicator (Covaris, Woburn, MA, USA), and libraries were constructed using the SureSelect XT Human All Exon v5 kit (Agilent Technologies, Foster City, CA, USA). This kit was designed to enrich 336,000 exons from 21,000 genes, covering 71 Mb of the genome. Library preparation included gDNA shearing, end-repair, A-tailing, adapter ligation, and sequencing in 100 bp paired-end mode on a HiSeq 2500 platform (Illumina Inc., San Diego, CA, USA). Library quality and quantity were assessed using a 2200 TapeStation and Qubit 2.0 fluorometer.

Sequencing reads were aligned to the UCSC hg19 reference genome (http://genome.ucsc.edu) using Burrows-Wheeler Aligner (v0.6.2) (https://bio-bwa.sourceforge.net/). Data cleanup and PCR duplicate removal were performed with GATK-2.2.9 and Picard-tools-1.8 (http://broadinstitute.github.io/picard/). Point mutations were identified with MuTect (v 1.1.44) (https://github.com/broadinstitute/mutect). Variant annotation was performed using ANNOVAR (ver 2019Mar23) (https://annovar.openbioinformatics.org/).

### 2.5. Differentially Expressed Genes (DEG) Analysis

The DESeq2 R package (v1.42.1) was used to identify DEG between groups. A count matrix compatible with DESeq2 was generated using the DESeqDataSetFromMatrix function with raw read counts. Significant genes were selected after normalization, with thresholds set at |log_2_FoldChange| > 2, adjusted *p*-value (*padj*) < 0.00001. Results were visualized using the EnhancedVolcano R package (v1.20.0).

### 2.6. Immune Gene Sets

The immune gene set included the *IMMUNE_SYSTEM_PROCESS* gene set (MSigDB: M13664) and gene annotations from *GO:0002376* (immune system process). Using 332 immune-related gene sets, differentially expressed immune-related genes across groups were identified and analyzed with the limma R package (v3.58.1). Results were visualized using pheatmap function, tidyr package, and ComplexHeatmap function (for colorRamp2) in R.

### 2.7. Differentially Methylated Probes (DMP) Analysis

DMP were analyzed using the limma, minfi, missMethyl, and minfiData R packages. Raw IDAT files generated by the Illumina Infinium MethylationEPIC BeadChip platform (Illumina Inc., San Diego, CA, USA) were processed. Probe annotation was performed using IlluminaHumanMethylationEPICanno.ilm10b4.hg19 and IlluminaHumanMethylationEPICman-ifest. Significant probes were selected from the annotated data matrix using limma. *M*-values (logit-transformed *β*-values) were calculated, and DMP analysis was performed with thresholds of |log_2_FoldChange| > 2, and *padj* < 0.00001. Results were visualized using EnhancedVolcano.

### 2.8. Biological Network Analysis and Functional Analysis

The Cytoscape software tool (version 3.10.2, https://cytoscape.org/) was used to analyze the interaction networks between genes. The STRING app within Cytoscape was employed to perform protein–protein interaction (PPI) analysis using the STRING database. Network analysis confirmed the connection between genes, and hub genes within the gene set were identified. Each node in the network represents the protein encoded by a gene. The strength of interaction is indicated by edge thickness, while edge color corresponds to shared pathways, enabling interpretation. The STRING app is available at https://apps.cytoscape.org/apps/stringapp.

Pathways and molecular functions associated with significant genes identified from DEG and DMP analyses were characterized. Enrichment analysis was performed using Enrichr (http://amp.pharm.mssm.edu/Enrichr; 8 June 2023) to evaluate relevant biological pathways. Reactome Pathways 2024 data were used for the pathway analysis. Visualization and enrichment scores were generated using the ReactomePA R package (v1.46.0). Gene annotation was conducted using the org.Hs.eg.db R package (v3.18.0), with significance defined as *p*-value < 0.05.

### 2.9. Differentially Methylated Regions (DMR) Analysis

DMR were analyzed to assess methylation differences across groups using the DMRcate R package (v3.4.0). The EPICv1 array platform (Illumina) was used for DMP analysis. CpG sites were filtered at *FDR* < 0.05, identifying 144,798 CpG sites. Promoter regions (19,193) were annotated using the IRanges R package (2.36.0).

### 2.10. Single-Cell Sequencing Analysis

Single-cell RNA-seq data of hepatocellular carcinoma were downloaded from (https://www.ncbi.nlm.nih.gov/geo/query/acc.cgi?acc=GSE290925, 8 June 2023) and were analyzed using Seurat R package (v5.3.0). Cells were filtered based on nUMI ≥ 500, nGene ≥ 250, log_10_GenesPerUMI > 0.80, and mitoRatio < 0.20, resulting in a total of 140,501 cells included in the analysis. Gene expression was normalized using the LogNormalize method (scale factor = 10,000). The top 2000 highly variable genes were selected, and data scaling was performed using the ScaleData function. A Seurat object was created using the merge function, and the parameters assay.use = “RNA”, reduction = “harmony”, and dims = 1:20 were applied in the IntegrateLayers function to remove batch effects and improve consistency between samples.

### 2.11. Survival Analysis

To assess the clinical prognostic value of genetic markers significantly correlated with the tumor microenvironment in hepatocellular carcinoma, prognostic analysis was performed using the TCGA-LIHC dataset. Clinical data for TCGA-LIHC patients were retrieved via cBioPortal and processed using the TCGAbiolinks R package (v2.37.1) (https://github.com/BioinformaticsFMRP/TCGAbiolinks). This analysis included 298 HCC patients with disease-free survival (DFS) data. Patients were categorized into “high-expression” and “low-expression” groups based on the median gene expression level. DFS was compared between these two groups.

### 2.12. Statistical Analysis

Kaplan–Meier survival analysis was used to evaluate the correlation between gene expression levels and DFS. Differences in gene expression between groups were assessed using the independent *t*-test. Kaplan–Meier survival curves were plotted to visualize DFS, and statistical significance was determined using the log-rank test. All the plots were performed using ggplot2 package (v3.5.2). All statistical analyses were conducted using R software (v4.3.2), with statistical significance set at *p* < 0.05.

## 3. Results

### 3.1. Overall Workflow of TME-Driven Gene Identification with Hepatocellular Carcinoma Cohort

For the classification of hepatocellular carcinoma (HCC) patients (*n* = 60) according to differences in the tumor microenvironment (TME), immune cell enrichment scores were calculated using four cell type deconvolution algorithms (EPIC [12], xCell [13], MCP-counter [14], and quanTIseq [15]), and samples were classified into three groups. Using hierarchical clustering based on Euclidean distance, samples were divided into three groups according to immune activation status, demonstrating well-separated and cohesive clusters (Figure 1). Subsequently, differentially expressed genes between the two groups exhibiting the most distinct differences in immune activity, along with probes showing the greatest differences in methylation levels, were integrated. Among the significant genes (*n* = 69), the 10 genes with the highest correlation between expression and methylation levels were selected as candidate markers. The expression patterns of these candidates were then examined at the single-cell level. Finally, three gene markers were selected, and survival analysis was conducted to evaluate their association with clinical prognosis.

### 3.2. Classification and Characteristics of HCC Patients Based on TME

HCC patients were grouped into immune activation subtypes based on tumor microenvironment profiles inferred through deconvolution methods. Cluster 1 (*n* = 7), designated as the Immune-high group, exhibited high immune scores and strong immune cell activation. Cluster 2 (*n* = 27), the Immune-intermediate group, showed a mixture of immune cell activation and suppression, whereas Cluster 3 (*n* = 26), the Immune-low group, had low immune scores and minimal immune activation (Figure 2a).

Following classification, clinical characteristics of HCC cohort were compared across groups (Table 1, Appendix A). The mean age was 58.1 years (range, 42–76), and the gender distribution (male: 81.7% and female: 18.3%) showed no association with group classification. The mean tumor size was 4.4 ± 1.4 cm. The cohort included 46 patients with hepatitis B virus (HBV), 5 with hepatitis C virus (HCV), and 9 with alcoholic or non-alcoholic liver disease. Statistical analysis revealed no significant associations between these clinical characteristics and immune activation groups. To assess the robustness of the classification, multiple immune-related metrics were calculated: the cytolytic (CYT) score, reflecting CD8^+^ T cell and natural killer (NK) cell activity based on *GZMA* and *PRF1* expression; the gene expression profile (GEP) score; the ESTIMATE score; and the immune score representing T cell–related gene expression. Principal component analysis (PCA) further supported distinct clustering patterns (Figure 2b,c, Appendix A), with Cluster 1 demonstrating markedly higher immune activation and Cluster 3 showing the lowest immune activity. Genomic mutation profiling was conducted to investigate whether specific mutations contributed to the differences in immune activation (Figure 2d). Among the 30 most frequently mutated genes, *CTNNB1* and *TP53* each harbored mutations in 28% of patients (*n* = 17), ranking highest in frequency. Most variants were non-synonymous mutations, while stopgain or splice site mutations were detected predominantly in Cluster 2. Analysis of whole-exome sequencing data showed no detectable differences in mutation rates among the clusters. Tumor mutation burden (TMB) did not differ significantly among groups, indicating that overall mutation load was not directly associated with immune activation status (Appendix A).

### 3.3. Identification of Epigenetic–Transcriptomic Regulated Genes

To investigate gene expression differences between groups with distinct immune microenvironments, Cluster 1 (the group with the highest immune activity, designated as the inflamed group) and Cluster 3 (the group with the lowest immune activity, designated as the non-inflamed group) were compared. Differentially expressed gene (DEG) analysis was performed to identify genes specifically and highly expressed in each group (Figure 3a; log_2_FC > 2, *FDR* < 0.001). DEG analysis across the three clusters revealed that 436 genes were highly expressed in the inflamed group, whereas 177 genes were highly expressed in the non-inflamed group (Figure 3b). Pathway enrichment analysis was then conducted using the Reactome database to identify biological pathways influenced by these highly expressed genes (Figure 3c). In the inflamed group, pathways related to collagen formation, collagen degradation, extracellular matrix (ECM) degradation, and immune system function were enriched. In contrast, pathways associated with lipid metabolism and steroid metabolism were predominant in the non-inflamed group, clearly distinguishing the immune-active and immune-suppressed states. Next, we assessed the expression of immune-related genes between the two groups (Figure 3d). Expression of immune-related genes was markedly reduced in the immune-suppressed, non-inflamed group, whereas these genes were robustly expressed in the inflamed group. To identify genes highly expressed in both groups and potentially regulated by epigenetic modifications, differentially methylated probe (DMP) analysis was performed (Figure 3e,f; log_2_FC > 2, *FDR* < 0.001). Probes showing significant methylation differences were mapped to the corresponding genes, and for each gene the probe with the lowest *p*-value was selected. This analysis identified 69 genes whose expression levels were strongly influenced by DNA methylation in both groups, indicating tight epigenetic regulation (Figure 3f).

### 3.4. Epigenetic Regulation and Functional Characterization in Inflamed and Non-Inflamed HCC Groups

Twenty-six genes upregulated in the inflamed group were associated with hypomethylation, whereas 43 genes downregulated in the non-inflamed group were influenced by hypermethylation (Appendix A). Correlation analysis between gene expression levels and *β*-values revealed a clear distinction between the two groups, indicating that these 69 genes are closely related to differences in the HCC tumor microenvironment (Figure 4a). A network analysis was performed to identify the pathways in which these genes function, confirming protein–protein interactions (PPI) (Figure 4b). Genes in the inflamed group were primarily involved in immune-related functions, while genes in the non-inflamed group were enriched in pyrimidine catabolism, valine and leucine degradation, and amino acid metabolism, centered around the hub gene *AGXT2* (Appendix A). The 10 genes showing the highest correlations between expression levels and *β*-values were selected as candidate markers (Figure 4c). The absolute correlation coefficients (|r-values|) of these candidates were closest to 1.0, with *p*-values < 0.001. In the inflamed group, *ARL11* (cg01425731), *FAM113B* (cg25123566), *TNFAIP6* (cg25196374), *TNFSF8* (cg05185749), and *UBXN11* (cg12709196) showed the highest correlation values. In the non-inflamed group, *AGXT2* (cg16297030), *DPYS* (cg26109240), *KHK* (cg02586830), *SERPINC1* (cg02586830), and *SLC27A5* (cg0726085) exhibited the strongest correlations (Figure 4d). Differentially methylated region (DMR) analysis was conducted to identify methylation patterns affecting the expression of these markers (Figure 4e). Significant methylation differences were observed for *ARL11*, *TNFSF8*, *AGXT2*, *DPYS*, *SERPINC1*, and *SLC27A5*.

### 3.5. Cell Type Specificity and Prognostic Value of Key Epigenetic-Regulated Genes

To investigate the expression patterns of candidate genetic markers at the single-cell level, a single-cell transcriptome dataset (GSM8825751, comprising 12 samples) was utilized. Clustering analysis of 140,501 cells identified 21 distinct clusters, with T cells, NK cells, and hepatocytes constituting the largest proportions (Figure 5a,b; Appendix A). The expression of the ten candidate marker genes, selected from the inflamed and non-inflamed groups, was examined across these cell types (Figure 5c). As expected, *ARL11*, *FAM113B*, *TNFAIP6*, *TNFSF8*, and *UBXN11* were predominantly expressed in immune cells such as neutrophils, T cells, NK cells, macrophages, and monocytes. In contrast, *AGXT2*, *DPYS*, *KHK*, *SERPINC1*, and *SLC27A5* were mainly expressed in hepatocytes. Among these, *FAM113B* and *TNFSF8* showed broad and high expression across immune cells, while *AGXT2*, *DPYS*, and *SERPINC1* exhibited increased expression in hepatocytes (Figure 5d). Ultimately, *AGXT2*, *DPYS*, and *TNFSF8* were selected as HCC-specific epigenetic–transcriptomic gene markers that reflect significant differences in the tumor microenvironment, exhibiting distinct cell type specificity. To evaluate their clinical relevance, disease-free survival analysis was performed (Figure 5e). Higher expression of *TNFSF8*, predominantly expressed in the inflamed group, was associated with increased immune activity and improved patient prognosis (*p* = 0.037). Similarly, elevated expression of *AGXT2* and *DPYS*, primarily expressed in the non-inflamed group, correlated with better prognosis (*p* = 0.18 and *p* = 0.027, respectively). These findings suggest that enhanced pyrimidine catabolism and amino acid metabolism mediated by these genes contribute to tumor suppressive effects (Figure 3c and Figure 4b).

## 4. Discussion

This study aimed to investigate the tumor microenvironment (TME) and intratumoral heterogeneity of hepatocellular carcinoma (HCC) through comprehensive multi-omics analysis. The TME in HCC facilitates tumor proliferation, immune evasion, and drug resistance via complex interactions among cancer cells, immune cells, and stromal components. For instance, tumor-associated macrophages (TAMs), myeloid-derived suppressor cells (MDSCs), and regulatory T cells (Tregs) are known to contribute to immunosuppression and tumor immune escape mechanisms [16]. To better understand these processes, we employed a multi-layered approach utilizing multi-omics data to provide additional insight. We initially stratified the cohort based on immune activation profiles and characterized the distinct features of each subgroup. We further investigated epigenetic modifications underlying gene expression differences between these groups. Specifically, genes exhibiting inverse correlations between expression levels and promoter methylation were prioritized as candidate markers.

Based on these analyses, we identified several genes showing distinct expression and methylation patterns. Within the immune-active TME, *ARL11*, *FAM113B*, *TNFAIP6*, *TNFSF8*, and *UBXN11* were overexpressed in association with hypomethylation [17]. Among these, *TNFSF8* (Tumor Necrosis Factor Superfamily Member 8), a cytokine and CD30 ligand involved in immune cell signaling and T cell activation, was positively associated with favorable prognosis in HCC [18]. Upregulation of IFN-γ/Granzyme B enhances T cell responsiveness, while increased *TNFSF8* expression suppresses immune checkpoint molecules such as *PD-1* and *CTLA-4*, or counteracts their inhibitory effects, thereby preventing T cell exhaustion within the TME [19]. *TNFSF8* has also been implicated in immune activation in various cancers including lung cancer and hematologic malignancies [20]. Conversely, *TNFAIP6* has been reported to enhance glucose uptake and promote glycolysis in cancer cells, though its prognostic relevance in HCC remains unclear and warrants further investigation [21,22,23]. The functions of *ARL11*, *UBXN11*, and *FAM113B* are poorly characterized, underscoring the need for future mechanistic studies.

In contrast, in the immune-suppressed TME, increased expression of *AGXT2* has been shown to reduce intracellular cholesterol levels in HCC cells, downregulate LDL receptor (*LDLR*), and upregulate *PCSK9* expression [24,25]. Lowering cholesterol levels through increased expression of *AGXT2* exerts a beneficial effect on HCC, whereas elevated expression of *PCSK9* has an adverse influence. *PCSK9* promotes vascular invasion by inhibiting apoptosis. Accordingly, clinical studies investigating the therapeutic potential of *PCSK9* inhibitors to enhance treatment efficacy are actively underway across various carcinomas. This regulatory axis is clinically actionable, as *PCSK9* inhibitors such as alirocumab and evolocumab are approved for therapeutic use [26,27]. Notably, *AGXT2* and *DPYS* are involved in pyrimidine catabolism and are known markers of dihydropyrimidinase deficiency [28,29]. *SLC27A5* is a liver-enriched gene implicated in fatty acid and RNA metabolism, and its downregulation is associated with hepatic fibrosis, tumor aggressiveness, and recurrence. Meanwhile, *SERPINC1* is known to induce apoptosis in HCC cells, inhibit M2 macrophage polarization, and act as a tumor suppressor [30].

In our study, *CTNNB1* mutations were enriched in the Immune-low group, while *TP53* mutations were more frequent in the Immune-high group. Consistent with previous findings in HCC, *CTNNB1* mutations are associated with an “immune-excluded” phenotype characterized by reduced infiltration of CD4+, CD8+ T cells, NK cells, and macrophages, leading to poor response to immune checkpoint inhibitors [31,32]. However, the immune context of *TP53* mutant tumors may vary according to cancer type and tumor microenvironmental factors [33,34]. These complex immune landscapes suggest distinct roles for *CTNNB1* and *TP53* mutations in shaping the tumor immune microenvironment and emphasize the need for further validation in larger cohorts to clarify their predictive values in immunotherapy.

This study offers several potential clinical implications. First, given the limitations of single-omics approaches in capturing the complexity of the TME, our multi-omics strategy could inform the development of early diagnostic panels, such as liquid biopsy platforms that incorporate both gene expression and DNA methylation data. Second, understanding the epigenetic regulation of genes associated with immune activation and lipid metabolism may lead to new therapeutic targets—such as *PCSK9* inhibitors—to modulate gene activity. Third, therapeutic strategies could be devised to sustain high expression of favorable prognostic markers such as *TNFSF8*, *AGXT2*, and *DPYS*. These findings suggest that the coordinated regulation of immune and metabolic pathways may underlie the observed heterogeneity in HCC and highlight the potential of these gene markers as diagnostic and therapeutic targets.

Despite these insights, our study has several limitations. First, the cohort consisted of only patients with Edmondson-Steiner grade II HCC, which may reduce the generalizability of our findings to the broader HCC population. This design allowed for analytic consistency by minimizing histologic heterogeneity; however, future studies should include patients across a wider range of histologic grades to better capture disease diversity. Second, expanding the number of cell types using our own single-cell data will better represent all relevant populations. Lastly, further studies incorporating cell-based and animal model experiments will be essential to confirm the biological functions and therapeutic potential of these markers. Nonetheless, our integrative approach reveals key features of the immune and metabolic landscape in HCC and identifies epigenetically regulated gene markers that may serve as promising diagnostic or therapeutic targets with prognostic relevance.

## 5. Conclusions

Our integrative approach revealed key features of the immune and metabolic landscape in HCC and identified epigenetically regulated gene markers that may serve as promising diagnostic or therapeutic targets with prognostic relevance.

## Figures and Tables

**Figure 1 cancers-17-03565-f001:**
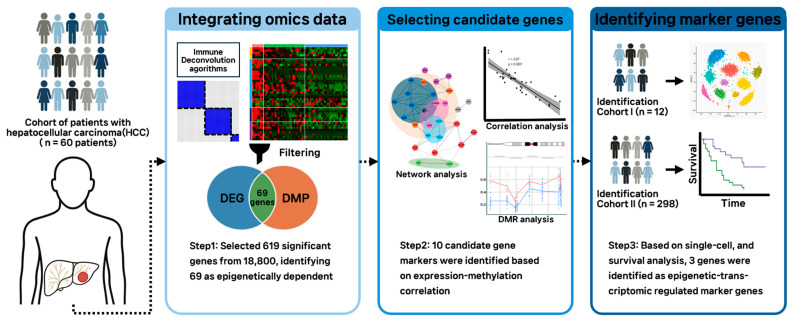
Analysis workflow of this study. A schematic diagram illustrating this study’s workflow including the integration of omics data, selection of candidate genes, and identification of marker genes.

**Figure 2 cancers-17-03565-f002:**
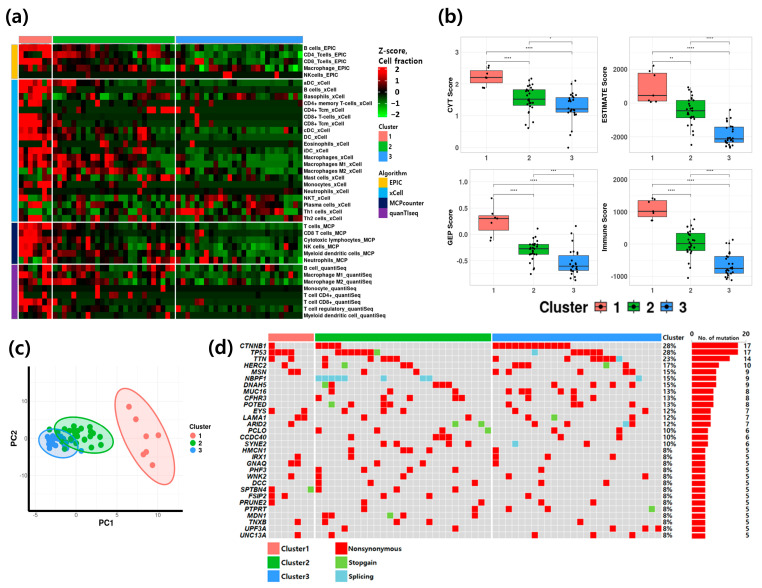
Comparison and characterization of sample groups stratified by immune microenvironment. (**a**) Complex heatmap illustrating the clustering results based on calculated immune scores. Sample-wise values were normalized using z-score scaling. * *p* < 0.05, ** *p* < 0.01, *** *p* < 0.001, **** *p* < 0.0001 (**b**) Boxplots showing differences in cytolytic activity (CYT) score, ESTIMATE score, gene expression profile (GEP) score, and immune score among the identified clusters. Statistically significant differences between groups were observed (*p*-value < 0.05 for all comparisons). (**c**) PCA plot illustrating the distribution of the 60 samples according to cluster assignment. (**d**) Oncoplot illustrating somatic mutations across clusters.

**Figure 3 cancers-17-03565-f003:**
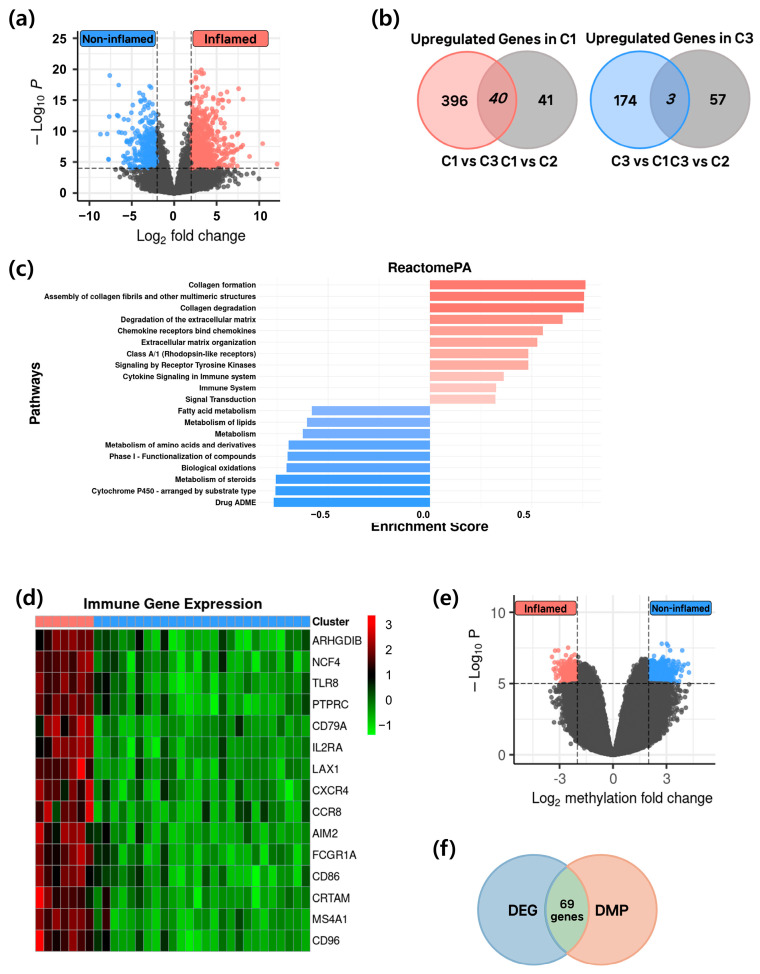
Differentially expressed genes regulated by epigenetic–transcriptomic mechanisms in inflamed and non-inflamed groups. (**a**) Volcano plot illustrating gene expression differences between inflamed group and non-inflamed group, which represent the most distinct immune environments (log_2_FC > 2, *FDR* < 0.001). (**b**) Venn diagram displaying the number of genes specifically expressed in each group. (**c**) Bar plot presenting pathway enrichment analysis of genes expressed in the inflamed and non-inflamed groups. The intensity of the bar color increases with higher enrichment scores. (**d**) Heatmap of expression patterns for immune gene sets in the two groups. (**e**) Volcano plot illustrating probes with the greatest differences in methylation levels between inflamed group and non-inflamed group, representing the most distinct immune environments (log_2_FC > 2, *FDR* < 0.001). (**f**) Venn diagram displaying the number of genes identified as significant in both DEG and DMP analyses.

**Figure 4 cancers-17-03565-f004:**
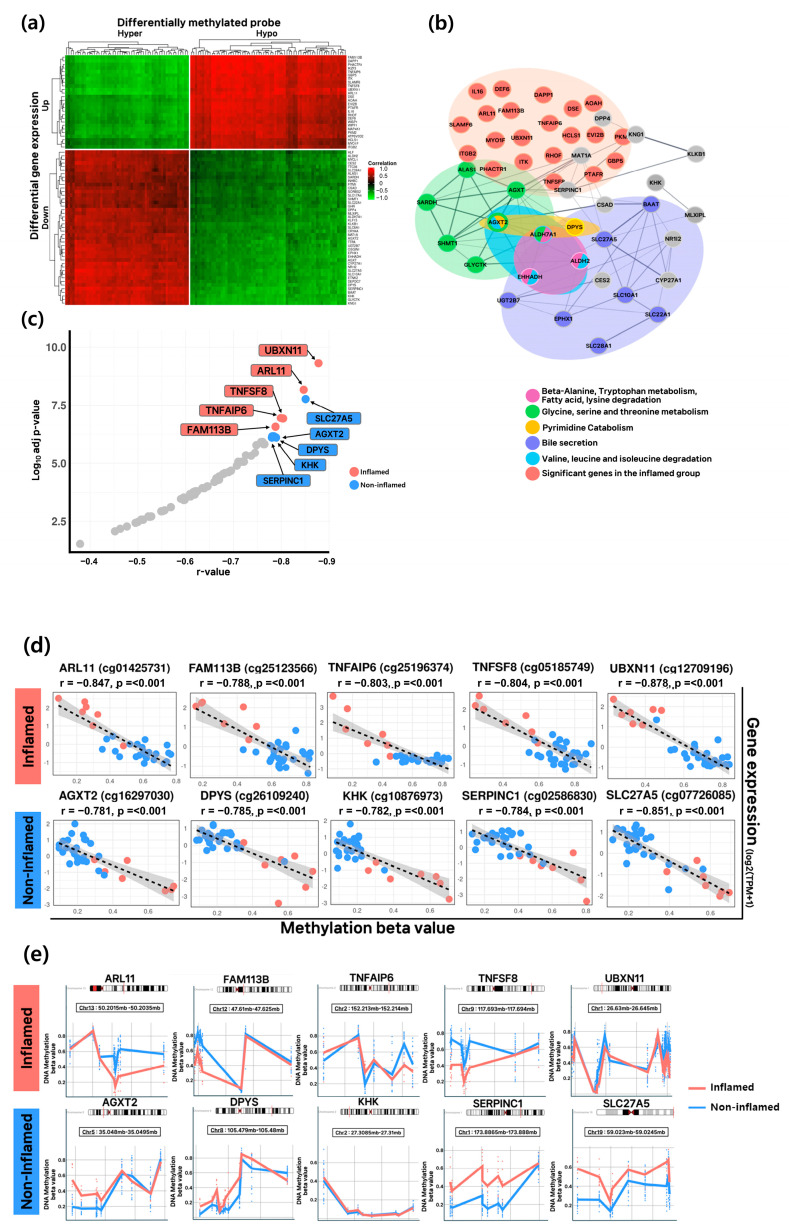
Selection of the most significant epigenetic–transcriptomic regulated genes between the inflamed and non-inflamed groups. (**a**) Heatmap illustrating the correlation between expression levels and *β*-values for significant genes identified in both DEG and DMP analyses. (**b**) Network analysis plot illustrating the functional roles of the 69 genes, with hub genes highlighted. (**c**) Scatter plot showing the negative correlation between gene expression and probe methylation. (**d**) Scatter plots illustrating the correlation between gene expression levels and *β*-values for the selected candidate marker genes across samples. (**e**) Plot illustrating the methylated regions of candidate marker genes, indicating the chromosomal locations and methylated CpG sites for each gene.

**Figure 5 cancers-17-03565-f005:**
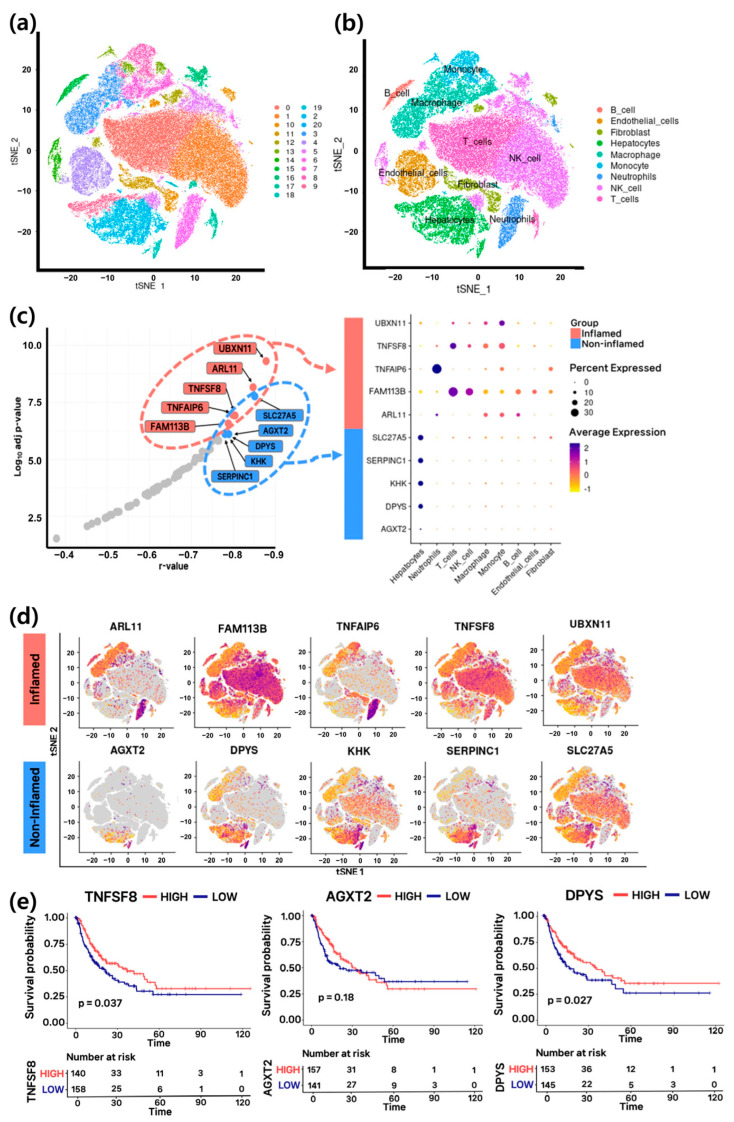
Expression patterns of candidate marker genes at the single-cell sequencing level. (**a**) tSNE plot of 12 samples, showing 21 distinct clusters. (**b**) tSNE plot illustrating the classification of nine cell types. (**c**) Dot plot showing the cell type–specific expression of candidate marker genes. (**d**) tSNE plot visualizing the expression of candidate marker genes. (**e**) Disease-free survival analysis using Kaplan–Meier curves for *TNFSF8*, *AGXT2*, and *DPYS*.

**Table 1 cancers-17-03565-t001:** The clinical characteristics of 60 patients with hepatocellular carcinoma.

Characteristics	Overall (n = 60)	Cluster 1 (n = 7)	Cluster 2 (n = 27)	Cluster 3 (n = 26)	*p*-Value
Age (range, years)	58.1 (42–76)	51.0 (42–61)	60.0 (44–76)	57.0 (43–73)	0.0629
Sex					0.6951
Male	49 (81.7%)	5 (71.4%)	23 (85.1%)	21 (80.7%)	
Female	11 (18.3%)	2 (28.6%)	4 (14.9%)	5 (19.3%)	
Tumor size (cm)	4.4 (±1.4)	5.3 (±1.5)	4.0 (±2.0)	4.5 (±2.0)	0.0767
Tumor SUV ^1^	4.5 (±2.6)	8.9 (±5.0)	3.7 (±1.1)	4.2 (±1.9)	0.023
Etiology					0.5355
HBV	46 (76.7%)	6 (85.7%)	22 (81.4%)	18 (69.2%)	
HCV	5 (8.3%)	1 (14.3%)	2 (7.4%)	2 (7.7%)	
Alcohol and others	9 (15%)	0 (0.0%)	3 (11.2%)	6 (23.1%)	
Cirrhosis					0.5923
Yes	22 (36.7%)	3 (42.8%)	8 (29.6%)	11 (42.3%)	
No	38 (63.3%)	4 (57.2%)	19 (70.4%)	15 (57.7%)	
Microvessel invasion					0.0275
Yes	30 (50.0%)	6 (85.7%)	9 (33.3%)	15 (57.7%)	
No	30 (50.0%)	1 (14.3%)	18 (66.7%)	11 (42.3%)	

^1^ SUV, standardized uptake value.

## Data Availability

The data that support the finding of this study are available from the corresponding author, upon reasonable request.

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
