# Peer review of "Integrated Multi-Omics Analysis Uncovers Immune–Metabolic Interplay in Hepatocellular Carcinoma Tumor Microenvironment"

_cancers, 2025, doi:10.3390/cancers17213565_

Round 1
Reviewer 1 Report
Comments and Suggestions for Authors
Manuscript ID: cancers-3915783
This study investigates the tumor microenvironment (TME) of hepatocellular carcinoma (HCC) through a multi-omics approach, analyzing transcriptomic, epigenomic, and single-cell RNA-sequencing data. It uncovers key immune-metabolic gene interactions, identifying epigenetically regulated markers like AGXT2, DPYS, and TNFSF8, which are linked to patient prognosis and could serve as potential therapeutic targets. The manuscript may be further improved by following suggestions.
- The word already mentioned in the title should not be repeated in keyword, select correct keyword relevant to the study.
- Use effect size for all statistical analyses to quantify the magnitude of differences or relationships, providing more meaningful insights beyond mere statistical significance.
- The study presents a novel multi-omics approach, but the sample size (60 patients) may limit the generalizability of the findings. Provide the equation/formula used for calculation of sample size?
- A more detailed discussion on the limitations of each data type would provide a clearer understanding of the methodology.
- The identification of epigenetically regulated genes is significant; however, further functional validation of these markers in vitro or in animal models is necessary to establish their clinical relevance.
- The choice of only HCC patients with Edmondson-Steiner grade II may introduce bias as including patients across various grades could provide a more comprehensive insight into the disease's progression.
- Single-cell RNA sequencing adds valuable resolution, but the number of cell types analyzed could be expanded to ensure all relevant populations are adequately represented.
- While survival analysis provides interesting insights, the clinical applicability of the markers identified needs further exploration in larger, independent cohorts.
- The role of immune cells in HCC's TME is highlighted well, but a deeper mechanistic explanation of how these cells interact with the identified metabolic pathways would strengthen the conclusions.
- Study lacks a discussion on the PCSK9 inhibitors current clinical trials or applicability in HCC treatment.
- The pathway analysis could be more detailed by including additional signaling pathways that might be relevant to HCC's metabolic and immune dysfunction.
- Future studies should investigate the dynamic nature of the TME in response to therapies, exploring how the identified biomarkers could be monitored in real-time to track disease progression and treatment efficacy.
- How the findings of the present study are relevant on a global scale.
- Provide limitation of the study under separate heading may be after discussion part.
May be improved
Author Response
Thank you for the thoughtful comments and useful suggestions on our manuscript. We have revised the manuscript accordingly.
This study investigates the tumor microenvironment (TME) of hepatocellular carcinoma (HCC) through a multi-omics approach, analyzing transcriptomic, epigenomic, and single-cell RNA-sequencing data. It uncovers key immune-metabolic gene interactions, identifying epigenetically regulated markers like AGXT2, DPYS, and TNFSF8, which are linked to patient prognosis and could serve as potential therapeutic targets. The manuscript may be further improved by following suggestions.
Comments 1: The word already mentioned in the title should not be repeated in keyword, select correct keyword relevant to the study.
Response: Thanks for the thoughtful comment. We have revised the keyword list to exclude terms already mentioned in the title, ensuring that the selected keywords accurately represent the key aspects of our study: “Immune activity score; Epigenetic regulation; Transcriptome; Prognosis markers”.
Comments 2: Use effect size for all statistical analyses to quantify the magnitude of differences or relationships, providing more meaningful insights beyond mere statistical significance.
Response: We have provided effect sizes for the statistical analyses: 1) Effect sizes (r) for the clinical characteristics (shown in Table 1) are presented in Supplementary Table S1. 2) Effect sizes of the statistical tests for the CYT score, ESTIMATE score, GEP score, and immune score (shown in Figure 2b) are presented in Supplementary Table S2.
Comments 3: The study presents a novel multi-omics approach, but the sample size (60 patients) may limit the generalizability of the findings. Provide the equation/formula used for calculation of sample size?
Response: Based on the RnaSeqSampleSize R package, we calculated the minimum sample size required to detect differential gene expression between two groups. In our study, at least 38 samples were determined to be necessary for a valid comparison. Therefore, when performing pairwise comparisons among three groups, this sample size is considered sufficient. The parameters for the calculation are as follows: prior data indicate that the minimum average read count among the prognostic genes in the control group is 50, the maximum dispersion is 0.5, and the ratio of the geometric mean of normalization factors is 1. We assume a total of 21,000 genes tested, of which the top 500 genes are prognostic.
Comments 4: A more detailed discussion on the limitations of each data type would provide a clearer understanding of the methodology.
Response: Thanks for the thoughtful comment. Among the three data types, we did not detect any significant mutation patterns between the clusters using the whole-exome sequencing data. There may be differences related to other types of genomic events, such as noncoding or structural variations. A whole-genome analysis may be required in the future to investigate these differences. We have added this limitation to Section 3.2: “Analysis of whole-exome sequencing data showed no detectable differences in mutation rates among the clusters.” (Lines #266-267, 414-423)
Comments 5: The identification of epigenetically regulated genes is significant; however, further functional validation of these markers in vitro or in animal models is necessary to establish their clinical relevance.
Response: We fully agree with the reviewer that functional validation would enhance the clinical relevance of our findings. However, due to current resource and ethical constraints, in vitro and in vivo experiments are beyond the scope of the present study. Instead, we have strengthened the Discussion by incorporating supporting literature and have acknowledged this as a limitation, while outlining our plans for future functional validation:
“Lastly, further studies incorporating cell-based and animal model experiments will be essential to confirm the biological functions and therapeutic potential of these markers. Nonetheless, our integrative approach reveals key features of the immune and metabolic landscape in HCC and identifies epigenetically regulated gene markers that may serve as promising diagnostic or therapeutic targets with prognostic relevance.” (Lines #441-446)
Comments 6: The choice of only HCC patients with Edmondson-Steiner grade II may introduce bias as including patients across various grades could provide a more comprehensive insight into the disease's progression.
Response: We agree that including only grade II patients may limit the generalizability of our findings. To ensure analytic consistency, we focused on grade II cases, as stated in the manuscript. We acknowledge this limitation and plan to include a broader range of grades in future studies. We agree that restricting the study to grade II patients limits more comprehensive insight. To maintain analytic consistency, we focused on grade II but have documented this limitation and will expand to other grades in future research.
“First, the cohort consisted of only patients with Edmondson-Steiner grade II HCC, which may reduce the generalizability of our findings to the broader HCC population. This design allowed for analytic consistency by minimizing histologic heterogeneity; however, future studies should include patients across a wider range of histologic grades to better capture disease diversity.” (Lines #435-440)
Comments 7: Single-cell RNA sequencing adds valuable resolution, but the number of cell types analyzed could be expanded to ensure all relevant populations are adequately represented.
Response: Thank you for the helpful suggestion. We agree that expanding the number of cell types would better represent all relevant populations. For example, T cells can be divided into various subtypes, and hepatocytes can be further classified into hepatocytes and hepatoblasts to examine more specific functional marker characteristics. Single-cell RNA sequencing provides valuable resolution, and increasing cell type diversity in future studies will improve comprehensive analysis. This approach represents a promising direction for our future research.
“Second, expanding the number of cell types using our own single-cell data will better represent all relevant populations.” (Lines #440-441)
Comments 8: While survival analysis provides interesting insights, the clinical applicability of the markers identified needs further exploration in larger, independent cohorts.
Response: We have already analyzed survival correlations using a large cohort (TCGA-LIHC dataset). We also attempted to perform additional survival analyses with independent datasets. Unfortunately, no significant differences in prognosis were observed, possibly due to the small sample sizes of these datasets.
Comments 9: The role of immune cells in HCC's TME is highlighted well, but a deeper mechanistic explanation of how these cells interact with the identified metabolic pathways would strengthen the conclusions.
Response: Thank you for the insightful comment, which we initially overlooked. We have added a more detailed mechanistic explanation of the interactions between immune cells and the identified genes. In addition, we have specified the mechanisms of the three final gene markers in the Discussion section.
“Among these, TNFSF8 (Tumor Necrosis Factor Superfamily Member 8), a cytokine and CD30 ligand involved in immune cell signaling and T cell activation, was positively associated with favorable prognosis in HCC [18]. Upregulation of IFN-γ/Granzyme B enhances T cell responsiveness, while increased TNFSF8 expression suppresses immune checkpoint molecules such as PD-1 and CTLA-4, or counteracts their inhibitory effects, thereby preventing T cell exhaustion within the TME [19].” (Lines #388-393)
“Lowering cholesterol levels through increased expression of AGXT2 exerts a beneficial effect on HCC, whereas elevated expression of PCSK9 has an adverse influence. PCSK9 promotes vascular invasion by inhibiting apoptosis. Accordingly, clinical studies investigating the therapeutic potential of PCSK9 inhibitors to enhance treatment efficacy are actively underway across various carcinomas.” (Lines #401-406)
Comments 10: Study lacks a discussion on the PCSK9 inhibitors current clinical trials or applicability in HCC treatment.
Response: Thank you for the thoughtful comment. Recent evidence suggests that PCSK9 inhibition enhances anti-HCC immune responses and may synergize with PD-1 immunotherapy. (Lines #406–407)
Comments 11: The pathway analysis could be more detailed by including additional signaling pathways that might be relevant to HCC's metabolic and immune dysfunction.
Response: We have included a list of signaling pathways associated with metabolic reprogramming and immune dysregulation in Supplementary Table S1.
Comments 12: Future studies should investigate the dynamic nature of the TME in response to therapies, exploring how the identified biomarkers could be monitored in real-time to track disease progression and treatment efficacy.
Response: We agree that investigating the dynamic changes in the tumor microenvironment (TME) during therapy is crucial. Our study identified key metabolic and immune biomarkers associated with distinct TME profiles and patient outcomes in HCC. Future studies will focus on real-time monitoring of these biomarkers to better track disease progression and treatment response using noninvasive detection methods such as ctDNA, exosomes, and circulating tumor cells (CTCs).
Comments 13: How the findings of the present study are relevant on a global scale.
Response: Hepatocellular carcinoma (HCC) remains a major global health burden and ranks among the most common cancers with high mortality worldwide. Our integrated multi-omics analysis, encompassing metabolic, epigenetic, and immune features of the tumor microenvironment, provides novel insights relevant to diverse HCC patient populations. The identification of metabolism- and immune-related genes with prognostic significance highlights molecular pathways that may be conserved across different geographic and etiologic contexts. Thus, our findings hold potential translational value for developing targeted therapies and prognostic markers on a global scale, complementing current efforts to reduce the worldwide burden of HCC.
Comments 14: Provide limitation of the study under separate heading may be after discussion part.
Response: We have comprehensively addressed the limitations of the study within the Discussion section. Therefore, a separate heading for the limitations after the Discussion was not included. (Lines #435-446)
Reviewer 2 Report
Comments and Suggestions for Authors
This is a well-structured and comprehensive study that successfully integrates transcriptomic, epigenomic, and single-cell RNA sequencing data to explore the immune–metabolic landscape of hepatocellular carcinoma (HCC). The methodological rigor, detailed workflow, and inclusion of validation analyses across multiple datasets significantly strengthen the credibility of your findings. The identification of AGXT2, DPYS, and TNFSF8 as epigenetically regulated markers associated with distinct immune and metabolic phenotypes offers valuable translational insight and may inform future therapeutic strategies.
While the manuscript is generally clear and well written, several minor improvements could enhance its impact. The Discussion section could be slightly condensed to emphasize the novelty and clinical implications of the identified gene markers, minimizing repetition of descriptive findings already covered in the Results. The Introduction may also benefit from a brief paragraph highlighting how this work differs from or expands upon prior multi-omics studies in HCC. Finally, a careful language and style revision is recommended to correct minor grammatical inconsistencies and ensure smoother transitions between sections.
Overall, this is a strong and technically sophisticated contribution to the understanding of HCC tumor biology. With these minor adjustments, the manuscript will be ready for publication.
Author Response
Thank you for the thoughtful comments and useful suggestions on our manuscript. We have revised the manuscript accordingly.
This is a well-structured and comprehensive study that successfully integrates transcriptomic, epigenomic, and single-cell RNA sequencing data to explore the immune–metabolic landscape of hepatocellular carcinoma (HCC). The methodological rigor, detailed workflow, and inclusion of validation analyses across multiple datasets significantly strengthen the credibility of your findings. The identification of AGXT2, DPYS, and TNFSF8 as epigenetically regulated markers associated with distinct immune and metabolic phenotypes offers valuable translational insight and may inform future therapeutic strategies.
Comments 1: While the manuscript is generally clear and well written, several minor improvements could enhance its impact. The Discussion section could be slightly condensed to emphasize the novelty and clinical implications of the identified gene markers, minimizing repetition of descriptive findings already covered in the Results.
Response: Thank you for the reviewer’s constructive comment. We have revised the Discussion section to be more concise, emphasizing the novelty and clinical implications of the identified gene markers while reducing repetition of previously described findings.
Comments 2: The Introduction may also benefit from a brief paragraph highlighting how this work differs from or expands upon prior multi-omics studies in HCC.
Response: Thank you for your valuable comment. We have added a paragraph in the Introduction section to clarify how our multi-omics approach differs from previous studies.
“Our study enhances prior HCC multi-omics research by integrating immune and metabolic features across genomic, transcriptomic, and epigenetic layers to delineate clinically relevant TME characteristics and identify potential therapeutic targets, whereas previous studies predominantly focused on molecular subtyping, mutation patterns, or immune profiling alone.” (Lines #73–77)
Comments 3: Finally, a careful language and style revision is recommended to correct minor grammatical inconsistencies and ensure smoother transitions between sections.
Response: We have revised the manuscript to correct minor grammatical inconsistencies and to improve the tone and flow, ensuring smoother transitions between sections.
Comments 4: Overall, this is a strong and technically sophisticated contribution to the understanding of HCC tumor biology. With these minor adjustments, the manuscript will be ready for publication.
Response: We really appreciate the reviewer’s comments. We have carefully addressed all suggested minor revisions, and we believe that these improvements have further enhanced the clarity and overall impact of the manuscript.
Reviewer 3 Report
Comments and Suggestions for Authors
1. The introduction provides a clear and logical overview of HCC. However, several sections in introduction could benefit from improved flow, conciseness, and avoidance of redundancy to enhance readability and precision.
2. The statement that “HCC is the most common solid malignancy worldwide” could be refined to clarify that it is the most common primary liver malignancy.
3. Please consider including drug-induced liver injury and chronic NAFLD abuse as emerging risk factors for HCC to provide a more comprehensive etiological overview.
4. Please specify total number HCC samples and normal tissue samples for clarity.
5. Justify the selection of only Edmondson-Steiner grade II tissues and its impact on generalizability.
6. Please indicate the RNA extraction kit or method and its reference or catalog number, as RNA isolation from tissue is a crucial step affecting transcriptome quality and reproducibility.
7. A brief description of library preparation, sequencing and IDAT files generation would improve clarity and reproducibility, helping readers understand key experimental steps and data processing.
8. Please provide a link or reference for the ConsensusClustarPlus R package (v1.66.0.) and any associated documentation or tutorials, so that readers can readily access the software for replication.
9. Please clarify the rationale for selecting three clusters in the consensus clustering.
10. Briefly describe any normalization and quality control steps applied to the raw counts.
11. Please clarify the rationale for the M-value threshold and significance cutoffs in the DMP analysis to justify the stringency of the results.
12. The biological or clinical significance of three final HCC gene markers should be elaborated to clarify their role in prognosis or TME modulation.
13. A brief summary interpreting how the identified immune subtypes relate to HCC prognosis and clinical outcomes would enhance the biological relevance of section 3.2.
14. In my view, the clinical relevance of section 3.2 can be improved by a brief explanation that links immune activation levels, important activation levels, important mutations (CTNB1, TP53),and their effects on prognosis or immunotherapy response.
Author Response
Thank you for the thoughtful comments and useful suggestions on our manuscript. We have revised the manuscript accordingly.
Comments 1: The introduction provides a clear and logical overview of HCC. However, several sections in introduction could benefit from improved flow, conciseness, and avoidance of redundancy to enhance readability and precision.
Response: Thank you for the valuable comment. We have revised the Introduction and Discussion sections to improve the flow and remove redundant content.
Comments 2: The statement that “HCC is the most common solid malignancy worldwide” could be refined to clarify that it is the most common primary liver malignancy.
Response: We appreciate the reviewer’s insightful comment. We have revised the sentence as follows: “Hepatocellular carcinoma (HCC) is the most common primary liver malignancy worldwide.” (Line #43)
Comments 3: Please consider including drug-induced liver injury and chronic NAFLD abuse as emerging risk factors for HCC to provide a more comprehensive etiological overview.
Response: Thank you for the helpful comment. We have revised the etiological risk factors of HCC as follows: “The major etiologies of HCC include chronic infection with hepatitis B or C virus, alcoholic liver disease, and metabolic disorders such as NAFLD and NASH. Moreover, drug-induced liver injury (DILI) and chronic metabolic dysfunction–associated liver disease are recognized as growing contributors to HCC development.” (Lines #45–49)
Comments 4: Please specify total number HCC samples and normal tissue samples for clarity.
Response: Thank you for the detailed comment. We have added the information about 60 samples in Section 2.1, specifying both tumor and normal tissue samples. (Lines #87–88)
Comments 5: Justify the selection of only Edmondson-Steiner grade II tissues and its impact on generalizability.
Response: We agree that including only grade II patients may limit the generalizability of our findings. To ensure analytic consistency, we focused on grade II cases, as stated in the manuscript. We acknowledge this limitation and plan to include a broader range of grades in future studies. We agree that restricting the study to grade II patients limits more comprehensive insight. To maintain analytic consistency, we focused on grade II but have documented this limitation and will expand to other grades in future research.
“First, the cohort consisted of only patients with Edmondson-Steiner grade II HCC, which may reduce the generalizability of our findings to the broader HCC population. This design allowed for analytic consistency by minimizing histologic heterogeneity; however, future studies should include patients across a wider range of histologic grades to better capture disease diversity.” (Line #435-440)
Comments 6: Please indicate the RNA extraction kit or method and its reference or catalog number, as RNA isolation from tissue is a crucial step affecting transcriptome quality and reproducibility.
Response: Total RNA were extracted by purification with the Qiagen RNeasy Mini Kit (QIAGEN Inc, Valencia, CA. USA). We have added this information to the manuscript. (Lines #96–97).
Comments 7: A brief description of library preparation, sequencing and IDAT files generation would improve clarity and reproducibility, helping readers understand key experimental steps and data processing.
Response: Thank you for the insightful comment. We have added a description of the methylation array preparation workflow in Section 2.2.
“Genomic DNA samples were subjected to bisulfite conversion using the Zymo EZ DNA Methylation Kit according to the manufacturer’s instructions. The converted DNA was then amplified before hybridization to the Illumina Infinium MethylationEPIC BeadChip. Following hybridization, chips were stained and scanned to generate fluorescent signals, which were subsequently processed to produce IDAT files for downstream methylation analysis.” (Lines #107-112)
Comments 8: Please provide a link or reference for the ConsensusClustarPlus R package (v1.66.0.) and any associated documentation or tutorials, so that readers can readily access the software for replication.
Response: We have added the appropriate citation for the ConsensusClusterPlus R package. (Line #121).
Comments 9: Please clarify the rationale for selecting three clusters in the consensus clustering.
Response: We have clarified the rationale for dividing the samples into three clusters.
“Using hierarchical clustering based on Euclidean distance, samples were divided into three groups according to immune activation status, demonstrating well-separated and cohesive clusters (Fig. 1).” (Lines #218-220)
Comments 10: Briefly describe any normalization and quality control steps applied to the raw counts.
Response: Thank you for the insightful comment. Normalization of raw counts was performed using functions implemented in the DESeq2 package, and this was addressed in Section 2.5. (Lines #103-104)
Comments 11: Please clarify the rationale for the M-value threshold and significance cutoffs in the DMP analysis to justify the stringency of the results.
Response: We agree that the criteria used for DMP selection are very stringent. Specifically, we applied thresholds of |log₂FoldChange| > 2, and padj < 0.00001 in the analysis. Given that approximately 865,000 CpG sites were tested, using a conventional p-value cutoff of 0.05 would result in a large number of significant probes, including many false positives. Therefore, we adopted a more conservative cutoff to ensure the robustness and reliability of the identified DMPs."
Comments 12: The biological or clinical significance of three final HCC gene markers should be elaborated to clarify their role in prognosis or TME modulation.
Response: Thank you for the detailed comment. We have carefully considered this and added a comprehensive discussion on the significance of the final gene markers in the Discussion section (Lines #385–413), elaborating on their roles in prognosis and tumor microenvironment modulation.
Comments 13: A brief summary interpreting how the identified immune subtypes relate to HCC prognosis and clinical outcomes would enhance the biological relevance of section 3.2.
Response: We performed survival analyses across the three immune subtype clusters. Although no significant differences in progression-free survival were observed among the subtypes (p = 0.55), a more focused, subtype-specific analysis of patients receiving immunotherapy may reveal clinically meaningful associations.
Comments 14: In my view, the clinical relevance of section 3.2 can be improved by a brief explanation that links immune activation levels, important activation levels, important mutations (CTNB1, TP53),and their effects on prognosis or immunotherapy response.
Response: Thank you for the thoughtful comments. In our study, CTNNB1 mutations were enriched in the Immune-low group, while TP53 mutations were more frequent in the Immune-high group. Consistent with previous findings in HCC, CTNNB1 mutations are associated with an "immune-excluded" phenotype characterized by reduced infiltration of CD4+, CD8+ T cells, NK cells, and macrophages, leading to poor responses to immune checkpoint inhibitors [31, 32]. However, the immune context of TP53 mutant tumors may vary according to cancer type and tumor microenvironmental factors [33, 34]. These complex immune landscapes suggest distinct roles for CTNNB1 and TP53 mutations in shaping the tumor immune microenvironment and emphasize the need for further validation in larger cohorts to clarify their predictive values in immunotherapy. (Lines #414- 423)
- Xiao, X., H. Mo, and K. Tu, CTNNB1 mutation suppresses infiltration of immune cells in hepatocellular carcinoma through miRNA-mediated regulation of chemokine expression. Int Immunopharmacol, 2020. 89(Pt A): p. 107043.
- Luke, J.J., et al., WNT/beta-catenin Pathway Activation Correlates with Immune Exclusion across Human Cancers. Clin Cancer Res, 2019. 25(10): p. 3074–3083.
- Long, J., et al., Development and validation of a TP53-associated immune prognostic model for hepatocellular carcinoma. EBioMedicine, 2019. 42: p. 363–374.
- Vadakekolathu, J., et al., TP53 abnormalities correlate with immune infiltration and associate with response to flotetuzumab immunotherapy in AML. Blood Adv, 2020. 4(20): p. 5011–5024.
Round 2
Reviewer 3 Report
Comments and Suggestions for Authors
I have studied the revised manuscript and response file. I found that the authors sufficiently revised the manuscript. In addition, the authors have provided a nicely detailed and thorough response to the comments from the previous review and have addressed my concerns. In my view, this manuscript can be published in the present revised form.